# Impact of Systemic Autoimmune Diseases on Treatment Outcomes and Radiation Toxicities in Patients with Stage I Non-Small Cell Lung Cancer Receiving Stereotactic Body Radiation Therapy: A Matched Case-Control Analysis

**DOI:** 10.3390/cancers14235915

**Published:** 2022-11-30

**Authors:** Zhe Chen, Hotaka Nonaka, Hiroshi Onishi, Eiji Nakatani, Mitsuhiko Oguri, Masahide Saito, Shinichi Aoki, Kan Marino, Takafumi Komiyama, Kengo Kuriyama, Masayuki Araya, Licht Tominaga, Ryo Saito, Yoshiyasu Maehata, Ryoji Shinohara

**Affiliations:** 1Department of Radiology, Shizuoka General Hospital, Shizuoka 420-8527, Japan; 2Department of Radiology, School of Medicine, University of Yamanashi, Chuo 409-3898, Japan; 3Department of Radiology, Fuji City General Hospital, Fuji 417-8567, Japan; 4Graduate School of Public Health (Medical Statistics), Shizuoka Graduate University of Public Health, Shizuoka 420-0881, Japan; 5Proton Therapy Center, Aizawa Hospital, Matsumoto 390-8510, Japan; 6Department of Radiology, Toranomon Hospital, Tokyo 105-8470, Japan; 7Department of Radiology, Shimada Municipal Hospital, Shimada 427-8502, Japan; 8Department of Radiology, Yamanashi Prefectural Hospital, Kofu 400-8506, Japan; 9Department of Health Sciences, Basic Science for Clinical Medicine, University of Yamanashi, Chuo 409-3898, Japan

**Keywords:** lung cancer, systemic autoimmune disease, stereotactic body radiation therapy, prognosis prediction, radiation pneumonitis

## Abstract

**Simple Summary:**

Lung cancer is the leading cause of cancer-related deaths worldwide. Stereotactic body radiation therapy (SBRT) has become the standard treatment for inoperable early stage non-small-cell lung cancer (NSCLC). This study explored the relationship between systemic autoimmune diseases (SADs) and SBRT in patients with stage I NSCLC, while eliminating the effects of clinical staging, irradiation techniques, and combination therapies. Our findings indicated that compared to controls, patients with NSCLC and SADs experience poorer prognosis, but an equal incidence rate of radiation pneumonitis, after SBRT. Therefore, we suggest that SBRT should be considered a radical treatment in patients with stage I NSCLC accompanying SADs; however, practitioners must keep the associated poor prognosis in mind.

**Abstract:**

We aimed to evaluate the impact of systemic autoimmune diseases (SADs) on treatment outcomes and radiation toxicities following stereotactic body radiation therapy (SBRT) for stage I non-small cell lung cancer (NSCLC). We queried an institution-based database on patients with SADs treated with SBRT for lung cancer between 2001 and 2016 (SAD group). Each patient was matched to three controls without SADs. The primary outcomes of interest were the overall survival (OS) and local control rate (LCR). The secondary outcomes were radiation toxicities of grades ≥2 (≥G2). Twelve patients with SADs were matched to 36 controls. The median follow-up duration was 3.6 years. There was a significant intergroup difference in the OS (hazard ratio [HR]: 4.11, 95% confidence incidence [CI]: 1.82–9.27, *p* < 0.001) and LCR (HR: 15.97, 95% CI: 2.89–88.29, *p* < 0.001). However, there were no significant intergroup differences in the odds of acute (odds ratio [OR]: 0.38, 95% CI: 0.02–8.91, *p* = 0.550) and late (OR: 2.20, 95% CI: 0.32–15.10, *p* = 0.422) ≥G2 radiation pneumonitis. No other ≥G2 toxicities were identified. In conclusion, although radiation toxicities are not enhanced by SADs, SADs are risk factors of poor prognosis following SBRT for stage I NSCLC.

## 1. Introduction

Lung cancer is the second-most commonly diagnosed cancer worldwide and is the leading cause of cancer-related deaths [1]. Non-small cell lung cancer (NSCLC) is defined as any pathological type of lung cancer, except small-cell cancer, and accounts for approximately 80–85% of all lung cancer cases [2,3].

Radiation therapy (RT) has contributed to radical and palliative treatments for patients with lung cancer. Recently, Stereotactic body RT (SBRT) has developed remarkably, and studies have revealed excellent local control and tolerable toxicity following their administration in patients with stage I NSCLC. Furthermore, SBRT has become the preferred treatment option for medically inoperable patients with significant comorbidities and for those who decline surgery [4].

Systemic autoimmune diseases (SADs) are heterogeneous disorders that are caused by immune system dysregulation, which leads to the activation of immune cells against autoantigens and inappropriate inflammation and multi-tissue damage [5,6]. Approximately 14–25% of patients with lung cancer have SADs [7,8]. However, due to safety concerns, such as RT might trigger the onset of SADs, practitioners are hesitant while offering RT to patients with SADs [9,10]. Most clinical trials on RT for patients with lung cancer also have excluded those with SADs. Severe radiation toxicity in a patient with SAD was initially reported in 1967 [11]. Only few studies on this disorder have been reported, and very few have described the use of RT for patients with NSCLC and SADs. Furthermore, most of these studies have focused on patients with SADs involving the connective tissue [12,13,14]. Moreover, the analyses in these studies were not controlled for clinical staging, combination therapies, and irradiation techniques. Currently, the effect of SADs ack on patients with NSCLC is unknown. Hence, we aimed to assess the relationship between the two, focusing specifically on SBRT for stage I NSCLC to avoid the effect of other variables.

## 2. Patients and Methods

### 2.1. Study Patients

This study was approved by the institutional review board of the University of Yamanashi on 16 February 2017 (application number: 1582). The requirement of written informed consent was waived due to the study’s retrospective observational design. The medical records of 327 patients with NSCLC who were treated with SBRT between June 2001 and December 2016 were reviewed retrospectively for a diagnosis of an SAD. The inclusion criteria were as follows: (1) diagnosis of primary NSCLC, (2) clinical stage I (T1a-T1bN0M0 and T2aN0M0) based on the TNM classification (seventh edition) [15], (3) diagnosis of SAD, and (4) availability of detailed patient information. Meanwhile, patients diagnosed with SAD after completion of RT were excluded. A 1:3 match was attempted between patients with SADs (SAD group) and those without SADs (control group) using the following match criteria: age, sex, performance status (PS), T stage, type of pathology, and total radiation dose.

### 2.2. RT Treatments

SBRT was performed with a linear accelerator using multiple noncoplanar static ports or dynamic arcs. Kilovoltage cone-beam computed tomography (CT) was used for imaging guidance during each treatment session. Respiratory motion management involved a self-breath-holding technique with a respiration-monitoring device [16]. The prescription doses and fractionations were as follows: 2001–2004, 60 Gy/10 Fr for T1 or 70 Gy/10 Fr for T2; 2005–2010, 48 Gy/4 Fr for T1 and T2; and 2010–2016, 50 Gy/4 Fr for T1 or 55 Gy/4 Fr for T2. To meet the dose constraints imposed in 2005–2016, 60 Gy/10 Fr or 70 Gy/10 Fr was adopted as the dose in cases where a tumor was located close to an organ at risk, such as the heart, trachea and primary bronchus.

### 2.3. Follow-Up

After completion of SBRT, CT was generally performed every 3 months in the first year, every 3–6 months in the second year, and every 6–12 months thereafter to up to 5 years or until death, regardless of the presence of progressive disease. If CT indicated the presence of progressive disease (recurrence or metastasis), fluorodeoxyglucose-positron emission tomography/CT was performed.

### 2.4. Toxicities

Acute toxicity occurring during RT and within 90 days of treatment completion was graded according to the Common Terminology Criteria for Adverse Events (CTCAE), version 4.0. These criteria were also adopted for grading late toxicity, which was defined as toxicity occurring after 90 days following treatment completion. Acute toxicity included radiation dermatitis, radiation pneumonitis (RP), and pain; late toxicity included RP, radiation-induced heart disease (RIHD), rib fracture, and pain.

### 2.5. Evaluation and Statistics

The disease and patient characteristics are presented using descriptive statistics, such as mean ± standard deviation or median (range) for continuous variables and number (percentage) for categorical variables. All continuous outcome variables were checked for normality and homogeneity of variance before the statistical analyses. Continuous and categorical data on patient characteristics were compared between the SAD and control groups using an unpaired Student’s t-test (Wilcoxon test for nonnormal data) and a chi-squared test, respectively.

The primary outcomes of interest were the overall survival (OS) and local control rate (LCR). OS was calculated from the date of SBRT completion to the date of death, or the last follow-up date. LCR was calculated to the first local recurrence date, censored death, or the last follow-up date. OS and LCR were estimated using the Kaplan–Meier method, and the survival difference between the groups was assessed using the log-rank test. No multivariable analysis was performed because all covariates were used for matching. In addition to the OS and LCR, acute and late toxicities of grade 2 or worse (≥G2) were compared between patients and controls using logistic regression. For tables with zero cell counts, logistic regression analysis was performed with the Firth correction. All statistical analyses were performed using the R statistical package (R for Macintosh version 4.2.1; The R Foundation for Statistical Computing, Vienna, Austria). *p* < 0.05 was considered statistically significant.

## 3. Results

### 3.1. Patients

Twelve patients with documented SADs at the time of RT were matched with 36 controls (each patient was matched with three controls); the demographic and clinical characteristics are listed in Table 1. The median dose of RT was 50 Gy (48–70 Gy). There were no significant differences in the patient characteristics and treatments between patients with SADs and the controls. Following a pathological examination, 23, 13, and 12 patients had adenocarcinomas, squamous cell carcinomas, and other pathologically confirmed tumors (including large cell carcinomas, spindle cell carcinomas, and not otherwise specified NSCLC), respectively. The T stage was T1 and T2a in 34 and 14 patients, respectively. Five patients showed central type lung cancer, and none of them were in the SAD group. Among the 12 patients who had SADs, five, three, and two had rheumatoid arthritis (RA), membranous nephropathy, and microscopic polyangiitis, respectively. The following SADs each affected one patient: primary biliary cholangitis and myeloperoxidase-anti-neutrophil cytoplasmic antibody-associated nephritis. Among the 12 patients with SADs, four patients (33.3%) were administered disease-modifying antirheumatic drugs (DMARDs) and eight (66.7%) patients were administered systemic steroids before SBRT. Detailed information on SAD patients is presented in Table 2.

### 3.2. Clinical Outcomes

The median follow-up duration was 3.6 years (range, 0.1–10.4 years). A total of 32 patients died from any cause during the study period, and eight patients experienced local recurrence. The clinical outcomes are summarized in Table 3. The median OS in the SAD and control groups was 2.7 and 7.9 months, respectively (hazard ratio (HR): 4.11, 95% confidence incidence (CI): 1.82–9.27, *p* < 0.001; Figure 1). The median LCR was 2.3 years in the SAD group and not reached in the control group (HR: 15.97, 95% CI: 2.89–88.29, *p* < 0.001; Figure 2), respectively. 

### 3.3. Toxicities

Most observed acute toxicities were RP and radiation dermatitis, although radiation dermatitis beyond G1 was not observed. Fatigue, vomiting and pain were not commonly observed. RP was the most commonly observed late toxicity. There was no case of RIHD, rib fracture, and chest pain in this cohort.

RP was the only ≥G2 toxicity that was observed throughout the treatment period; therefore, RP was evaluated. Among the 12 patients with SADs, none developed acute ≥G2 RP but two developed late ≥G2 RP. Of these two patients, one died of G5 RP and another developed G3 RP. Meanwhile, among the controls, three developed acute ≥G2 RP and another three developed late ≥G2 RP. Of the three patients who developed early ≥G2 RP, two developed G2 RP and another developed G3 RP. Meanwhile, of the three patients who developed late ≥G2 RP, one developed G4 RP and another developed G3 RP. There was no significant difference in the incidence of acute (odds ratio (OR): 0.38, 95% CI: 0.02–8.91, *p* = 0.550) and late (OR: 2.20, 95% CI: 0.32–15.10, *p* = 0.422) ≥G2 RP between the SAD and control groups (Table 4).

## 4. Discussion

SBRT has become the best alternative to surgery as a valid treatment option for medically inoperable patients with stage I NSCLC [17]. To date, only few studies have systematically investigated the direct relationship between SADs and RT for lung cancer; however, these have obtained conflicting findings [12,18]. To the best of our knowledge, this is the first report on a relationship between the two, focusing on SBRT for stage I NSCLC to eliminate the effects of clinical staging, irradiation techniques, and combination therapies. Our findings demonstrate that patients with SAD have poor OS and LCR following SBRT for stage I NSCLC.

No major landmark trials have established the efficacy and safety of SBRT for patients with stage I NSCLC and SADs. Shaikh et al. performed a meta-analysis of 10 studies (*n* = 4028 patients) and reported no significant differences in the OS, loco-regional recurrence-free survival, and distant metastasis-free survival between patients with collagen vascular disease (CVD) and the controls [19]. However, that study analyzed too many RT techniques and treatment sites. Diao et al. compared the efficacy and toxicity of RT between 31 patients with intrathoracic malignancies (including 29 with lung cancer) who had a history of CVD and 856 controls [12]; theirs is the only detailed report on the systematic assessment of the relationship between RT and lung cancer in patients with CVD. All patients were treated with three-dimensional conformal RT or intensity-modulated RT (none were treated with SBRT). Approximately 80% of the patients had stage III or IV cancer and underwent concurrent chemotherapy; no significant differences in the OS or other cancer-related outcomes were observed between the study and control groups. However, our study revealed a significantly poorer OS after SBRT in the SAD group than in the control group. Furthermore, Cox regression analysis implied that the HR of death had increased by approximately four-fold every year in the SAD group. In our study, controlling for lung cancer stages and irradiation techniques and excluding patients who received combination therapy may have caused the statistically significant between-group differences.

Our findings also revealed a poorer LCR in the SAD group. However, Diao et al. and Shaikh et al. reported no significant differences in the loco-regional recurrence-free survival between patients with CVD and the controls [19]. This difference may be due to the aforementioned reasons and our study’s homogeneous patient population. Certain factors may contribute to poor local recurrence in the SAD group, either alone or in combination with others. First, patients with SAD demonstrate an insufficient immune response due to activation of autoimmunity, and the antitumor immune response could have been lowered. Second, irradiation activates an immune response; irradiated tumor cells release proteins (such as immunostimulants or tumor antigens) that are processed by antigen-presenting cells. Subsequently, tumor-specific cytotoxic T lymphocytes are activated, which attack existing and underlying tumor cells throughout the body [20,21,22,23].

Previous studies revealed that a high proportion of patients with CVD who are irradiated to the breast, pelvis, or abdomen are associated with higher rates of radiation-induced toxicity [19,24]. Diao et al. reported that patients with CVD had a significantly increased risk of RP after RT [12]. However, two matched-control studies independently reported no differences in the acute or late toxicities between patients with and without CVD [25,26]. Morris et al. reported that the risk of late toxicity in patients with RA was not superior to that in historical controls. They suggested that the incidence and severity of radiation toxicity may vary with the treatment site [24]. Shaikh et al. also concluded that irradiated to the thorax had significantly higher rates of late G2/3+ radiation toxicity [19]. However, our findings indicated that there was no significant increase in the incidence of acute and late ≥G2 RP between the SAD and control groups, and toxicities should not be considered contraindications to SBRT.

Recently, the role of RIHD in patients with NSCLC has emerged as a topic of interest [27]. Previous studies have investigated the risk of cardiac toxicity and the effect of the cardiac radiation dose on survival in patients with NSCLC [28,29,30]. However, most of these studies evaluated RIHD in patients with locally advanced NSCLC. Unlike conventionally fractionated RT, SBRT uses smaller treatment fields and involves a markedly different cardiac dosimetry, whereby the entire heart will be subject to lower mean doses [31]. When tumors are located near cardiac substructures in patients treated with SBRT, cardiac doses show large variability and depend on the tumor location [32]. There are two possible reasons for the low rate of RIHD in our cohort. First, modern SBRT planning methods may lead to the reduction of cardiac doses. We also reduced the dose per fraction to 6 Gy or 7 Gy in cases where the tumor was close to the cardiac substructures. Second, there were only five patients with central type lung cancer, and this small number may have influenced the incidence of RIHD. To the best of our knowledge, there is no relevant dose-volume recommendation for the heart in guidelines for SBRT. Lung cancer patients are generally older with more comorbidities; thus, all cardiac events have the potential to be clinically significant and life-threatening events. We suggest that cardiac doses should be limited as much as possible.

With advancements in irradiation technology and discoveries in radiobiology, practitioners are administering gradually increasing irradiation doses [33]. In this study, because radiation doses had an obvious correlation with the year of administration, we used the total radiation dose as a matching criterion rather than the irradiation year. Delivering a higher dose per fraction is known to increase the risk of damage to late-responding tissues. Our findings also revealed a trend of increasing acute and late toxicities in the SAD group; however, these changes were not significant. Lowell et al. reported no G3+ toxicities associated with gamma knife radiosurgery for intracranial tumors in patients with CVD [34]. Lin et al. conducted an international systematic review and meta-analysis on a modern series of contemporary RT techniques; they noted no specific associations between toxic effects and dose fractionation [35]. Although no firm conclusions can be drawn from these studies, available data have thus far indicated low rates of severe toxicity across various dose fractionation regimens and are consistent with the findings from our study. Therefore, we think a broader use of SBRT for the thorax is possible after the recent advancements in treatment techniques and imaging guidance.

This study has several limitations. First, was the retrospective study design with a small sample size. Although our findings were statistically significant, the effect size was small. Our objective was to demonstrate that SAD will affect survival. Although SAD was a negative predictor of OS and LCR in our cohort, the present study requires replication and validation with larger cohorts before the findings can be applied in clinical practice. Second, our analysis was focused on a group of SADs. Each SAD has a distinct natural history and relative radiosensitivity. DMARDs are also used variably depending on the symptoms, with some drugs having relatively more radiosensitizing effects than the others [36]. Third, driver gene mutations in patients with lung adenocarcinoma are closely related to the prognosis. However, given the retrospective design of this study, it was difficult to obtain mutation data for all patients. Therefore, the impact of gene mutation may not be fully considered. Finally, it has been reported that transplant recipients on immunosuppressants have an increased risk of cancer [37]. However, 92% of the patients with SADs in this study were treated with immunosuppressants (mainly prednisolone). There is no substantial evidence to suggest carcinogenicity of prednisolone [38]; therefore, it is not possible to clarify the relationship between immunosuppressants and prognosis.

## 5. Conclusions

In conclusion, although radiation toxicities are not enhanced by SADs, they are a risk factor for poor prognosis following SBRT for stage I NSCLC. Our findings suggest that SBRT should be considered a radical treatment in patients with stage I NSCLC accompanying SADs; however, practitioners must keep the associated poor prognosis in mind.

## Figures and Tables

**Figure 1 cancers-14-05915-f001:**
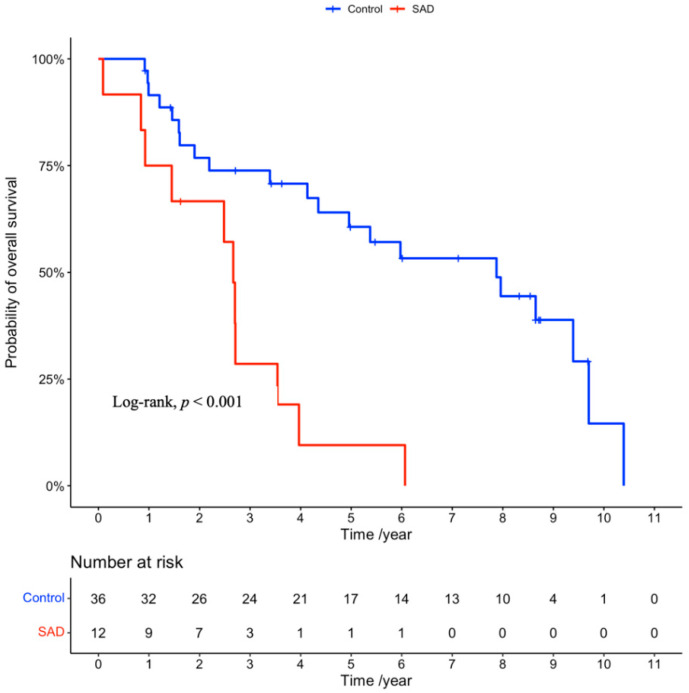
Kaplan–Meier curves for overall survival in the SAD and control groups. SAD: systemic autoimmune disease.

**Figure 2 cancers-14-05915-f002:**
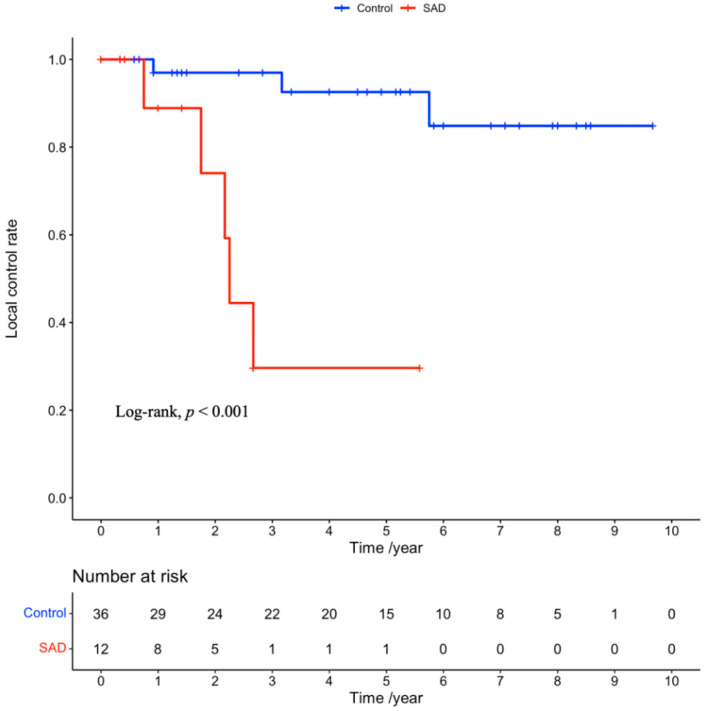
Kaplan–Meier curves for the local control rates in the SAD and control groups. SAD: systemic autoimmune disease.

**Table 1 cancers-14-05915-t001:** Patient characteristics.

Factor	Group	SAD Group	Control Group	*p*-Value
*n*		12	36	
Patients				
Age (years)	<79	7 (58.3)	24 (66.7)	0.731
	≥79	5 (41.7)	12 (33.3)	
Sex	Male	8 (66.7)	27 (75.0)	0.710
	Female	4 (33.3)	9 (25.0)	
KPS	≥80	12	36	-
Pathology	Adenocarcinoma	5	18	0.835
	Squamous cell carcinoma	4	9	
	Other	3	9	
T stage	T1 (T1a/T1b)	7	27	0.294
	T2a	5	9	
Cancer location	Peripheral	12	31	0.312
	Central	0	5	
Treatments				
Prescription dose	48 Gy/4Fr	7	27	0.093
	50 Gy/4Fr	0	3	
	55 Gy/4Fr	1	3	
	60 Gy/10Fr	2	3	
	70 Gy/10Fr	2	3	

Abbreviations: SAD = systemic autoimmune disease, KPS = Karnofsky performance status, Fr = fraction.

**Table 2 cancers-14-05915-t002:** Detailed information for patients with systemic autoimmune diseases.

No.	Age	Sex	Pathology	T Stage	Location	Dose	Autoimmune Disease	Immunosuppressive Agent	Cancer Progression	Outcome
1	77	M	SCC	T2a	Right, peripheral	60 Gy/10Fr	MPO-ANCA associated nephritis	Prednisolone	Yes	Cancer-related death
2	72	M	AC	T2a	Right, peripheral	60 Gy/10Fr	Rheumatoid arthritis	Prednisolone	Yes	Cancer-related death
3	68	F	AC	T1a	Right, peripheral	48 Gy/12Fr	Rheumatoid arthritis	Prednisolone, methotrexate	Yes	Cancer-related death
4	66	M	AC	T1a	Right, peripheral	48 Gy/12Fr	Membranous nephropathy	Prednisolone	Yes	Death due to RP
5	67	M	SCC	T2a	Left, peripheral	48 Gy/12Fr	Membranous nephropathy	Prednisolone	No	Non-cancer-related death
6	82	M	SCC	T1b	Right, peripheral	48 Gy/12Fr	Rheumatoid arthritis	Prednisolone	Yes	Cancer-related death
7	77	M	AC	T1a	Right, peripheral	48 Gy/12Fr	Microscopic polyangiitis	Prednisolone	Ye	Non-cancer-related death
8	82	M	SCC	T1a	Right, peripheral	48 Gy/12Fr	Rheumatoid arthritis	Prednisolone	Yes	Non-cancer-related death
9	82	F	Other	T1a	Right, peripheral	48 Gy/12Fr	Rheumatoid arthritis	Methotrexate	No	Non-cancer-related death
10	80	F	Other	T2a	Right, peripheral	70 Gy/10Fr	Microscopic polyangiitis	Prednisolone	Yes	Cancer-related death
11	81	M	SCC	T2a	Right, peripheral	55 Gy/4Fr	Membranous nephropathy	Prednisolone, ciclosporin	Yes	Alive
12	65	F	AC	T1a	Right, peripheral	70 Gy/10Fr	Primary biliary cholangitis	N.A.	Yes	Cancer-related death

Abbreviation: F = female, M = male, Fr = fraction, SCC = squamous cell carcinoma, AC = adenocarcinoma, MPO-ANCA = myeloperoxidase-anti-neutrophil cytoplasmic antibody, RP = radiation pneumonia, N.A. = Not available.

**Table 3 cancers-14-05915-t003:** Survival outcomes.

Outcomes	SAD Group	Control Group	HR (95% CI)	*p*-Value
Overall survival rate			4.11 (1.82–9.27)	<0.001
3-year rate (%)	28.6	73.9		
5-year rate (%)	9.5	60.7		
Median (years)	2.7	7.9		
Local recurrence rate			15.97 (2.89–88.29)	<0.001
3-year rate (%)	70.4	3.0		
5-year rate (%)	100	7.4		
Median (years)	2.3	Not estimated		

Abbreviation: SAD = systemic autoimmune disease, HR = hazard ratio, CI = confidence interval.

**Table 4 cancers-14-05915-t004:** Rates of acute and late RP (≥G2) by the SAD and control groups.

Toxicities (%)	All Patients (*n* = 48)	SAD Group (*n* = 12)	Control Group (*n* = 36)	OR (95% CI)	*p*-Value
Acute				0.38 (0.02–8.91) *	0.550
<G2	45 (93.8)	12 (100)	33 (91.7)		
≥G2	3 (6.2)	0 (0)	3 (6.2)		
Late				2.20 (0.32–15.10)	0.422
<G2	43 (89.6)	10 (83.3)	33 (91.7)		
≥G2	5 (10.4)	2 (16.7)	3 (8.3)		

Abbreviations: RP = radiation pneumonitis, G = grade, SAD = systemic autoimmune disease, OR = odds ratio, CI = confidence interval. * Firth correction was used for estimating the odds ratio, 95% CI, and *p*-value.

## Data Availability

The data presented in this study are available on request upon reasonable request from the corresponding author.

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
