# Peer review of "Impact of Systemic Autoimmune Diseases on Treatment Outcomes and Radiation Toxicities in Patients with Stage I Non-Small Cell Lung Cancer Receiving Stereotactic Body Radiation Therapy: A Matched Case-Control Analysis"

_cancers, 2022, doi:10.3390/cancers14235915_

Round 1

Reviewer 1 Report

This is an interesting study to deal with patients with stage I NSCLC and autoimmune diseases treated by SBRT. The strength is a study to deal with rare disease ( autoimmune diseases) and there have not been many publications regarding the toxicities and out come after SBRT for stage I NSCLC with autoimmune diseases treated by SBRT. The weakness of this paper are  a small number of patients(12 patients) and a retrospective study.

Author Response

Thank you for your valuable comments and insightful feedback. We hope this pioneer study brings this topic into focus. We have acknowledged the limitations of the retrospective study design and small sample size in the limitations paragraph.

Reviewer 2 Report

The authors aimed to evaluate the impact of systemic autoimmune diseases (SADs) on treatment outcomes and radiation toxicities following stereotactic body radiation therapy (SBRT) for stage I non-small cell lung cancer (NSCLC). The research is interesting and the result is sound, several major issues should be fixed before publication:

1. Since the number of participants is so small, the conclusion part should be more cautious. After all, a retrospective study of such a small sample cannot fully confirm any conclusion

2. The author can list the specific enrollment and outcomes of 12 SAD patients, which is more valuable for readers

3. The mutation of driving gene in patients with lung adenocarcinoma is closely related to the prognosis. Although it is difficult to obtain results in statistics, the main mutation should be carried out

Author Response

Thank you for your valuable comments and insightful feedback; we greatly appreciate your constructive suggestions. We agree with all of your comments and have rechecked and revised the manuscript accordingly. Please check our responses below.

  1. Thank you for your helpful feedback. We hope this pioneer study brings this topic into focus. We have revised the first limitation to highlight that although SAD was a negative predictor of OS and LCR in our cohort, the present study requires replication and validation with larger cohorts before the findings can be applied in clinical practice (Lines 286-290).
  2. Thank you for your constructive suggestion. We have inserted Table 2 to list the specific enrollment and outcomes of the 12 SAD patients (Lines 154-157).
  3. We completely agree that driver gene mutations in patients with lung adenocarcinoma are closely related to the prognosis. However, given the retrospective design of the study, we could not obtain mutation data for all patients. We rechecked our original data and found that mutation data were available only for patients treated after 2010. Thus, we could not obtain any statistical data. We have added this as a limitation of the study (Lines 294-297). We will consider driver gene mutations in our future studies.

Reviewer 3 Report

The authors present some very interesting findings.

1. Can the authors talk about why they did not include radiation induced heart disease or pericarditis as one of the adverse events when the role of radiation induced heart disease in patients with lung cancer is well established?

2. Can the authors talk about the impact of comorbidities in these patients?

Author Response

Thank you for your review and insightful feedback; we greatly appreciate your constructive suggestions. We agree with all of your comments and have rechecked and revised the manuscript accordingly. Please find our responses below.

  1. We apologize for the unclear description. We did not include radiation-induced heart disease (RIHD) because we did not find any case of RHID in this cohort.

Patients with locally advanced NSCLC are more likely to develop cardiac toxicities because irradiation of the heart is more common in these patients. SBRT can use smaller treatment fields to avoid the heart as much as possible. When we plan SBRT, we reduce the dose per fraction to 6 Gy or 7 Gy for cases where the tumor is located close to the cardiac substructures. We have clarified this in the manuscript (Line 98, line 142, lines 179-182, lines 254-270). Moreover, there were only five patients with central type lung cancer in this study, and this small number of patients may have influenced the incidence of RIHD. We also attempted a statistical analysis; however, we were unable to obtain any significant results.

  1. We agree that this point must be discussed in the manuscript. We have revised the Discussion in accordance with your suggestion (Lines 254-270).

Round 2

Reviewer 2 Report

The article could be accepted in present form.